# Analysis on Microbial Profiles & Components of Bile in Patients with Recurrent CBD Stones after Endoscopic CBD Stone Removal: A Preliminary Study

**DOI:** 10.3390/jcm10153303

**Published:** 2021-07-27

**Authors:** Jung Wan Choe, Jae Min Lee, Jong Jin Hyun, Hong Sik Lee

**Affiliations:** 1Department of Internal Medicine, Korea University Ansan Hospital, Ansan 15355, Korea; jwchoe@korea.ac.kr (J.W.C.); sean4h@korea.ac.kr (J.J.H.); 2Department of Internal Medicine, Korea University Anam Hospital, Seoul 02841, Korea; jmlee1202@gmail.com

**Keywords:** gallstones, bile, microbiota, bile acids and salts

## Abstract

Background/Aim: Common bile duct (CBD) stone recurrence after endoscopic treatment is a major concern as a late complication. Biliary bacterial factors and biochemical factors determine the path of gallstone formation. The aim of this preliminary study was to investigate the microbial profile and components of bile in patients with and without recurrent CBD stones after endoscopic CBD stone removal. Methods: Among patients who had undergone an initial endoscopic procedure for the removal of CBD stones and were followed up for >2 years, 11 patients who experienced at least two CBD stone recurrences, six months after endoscopic retrograde cholangiopancreatography (ERCP), were categorized into the recurrence group. Nine patients without CBD recurrence events were matched. Results: Polymicrobial infections are generally seen in all patients who have biliary sphincteroplasty. Microbial richness, measured by the numbers of operational taxonomic units (OTUs), was reduced in the recurrence group. The microbial evenness was also significantly lower than in the non-recurrence group. The overall microbial communities in the recurrence group deviated from the non-recurrence group. Infection with bacteria exhibiting β-glucuronidase activity was more frequent in the recurrence group, but there was no statistical significance. In an analysis of the bile components, the bile acid concentration was higher in the non-recurrence group than in the recurrence group. However, the other metabolites were not significantly different. Conclusions: Microbiota dysbiosis and altered bacterial community assembly in bile duct and decreased bile acid in bile juice were associated with recurrence of bile duct stone.

## 1. Introduction

Endoscopic stone removal by endoscopic retrograde cholangiopancreatography (ERCP) is the standard treatment for patients with cholangitis caused by common bile duct (CBD) stones. However, CBD stone recurrences after endoscopic stone removal is one of the most problematic late complications, with the rate ranging between 3% and 24% [1,2,3]. The recurred CBD stone could be a burden to both the individual’s health and the social healthcare industry.

Many factors have been suggested to be associated with CBD stone recurrence. These factors include abnormal bile duct anatomy (e.g., dilated CBD diameter or sharp angulation of the CBD), retrograde infection from the duodenum with anatomical changes (e.g., periampullary diverticulum (PAD) and manipulated ampulla), altered bile biochemical composition, etc. [3]. Among them, the anatomical risk factors for recurrent CBD stones cannot be changed, whereas biliary bacterial factors and biochemical factors, including bile composition, may be correctable.

Many studies have suggested that different bacterial species and bile composition may augment the formation of recurrent CBD stones [4,5,6]. Despite the importance of microbial inhabitants of human bile in the pathologic condition, our current knowledge is limited to a few species of culturable bacteria that have been associated with CBD stones. However, as multiple microorganisms coexist in the biliary tract, culture-dependent methods are somewhat insensitive and biased for bacterial identification and are inadequate to study the entire microbial community [7,8]. In this regard, our understanding on the exact contribution of unculturable or difficult to culture bacteria in recurrent CBD stone formation is very limited. Indeed, only a minor fraction (0.1 to 10%) of the bacteria can be cultivated using standard techniques [9]. Recently, the application of next-generation sequencing (NGS) has provided a more comprehensive understanding of the bacterial community and expanded the microbiota detected in humans [10]. Utilizing NGS technology, a total of 3.8 million bacterial species could be discriminated with 16S rRNA sequences discovered from previously collected data and published studies [11]. Unlike conventional methods for identifying microorganisms, NGS provides a comprehensive picture of the unculturable or difficult to culture bacteria by employing bacterial universal primers. The use of this method for the detection and differentiation of bacterial species could increase our understanding on the role of these bacteria and the manner of their pathogenesis in diseases. Nevertheless, there have been few such investigations that have looked into the biliary microbiome. In addition to the importance of microbiome in CBD stone recurrence, the role of bile composition has also not been fully elucidated.

Therefore, this study was carried out to investigate the bacterial communities in bile using NGS and to analyze the metabolic components of bile to elucidate the role of microbiota and bile components in CBD stone recurrence.

## 2. Materials and Methods

### 2.1. Patients and Bile Sample Collection

Patients who have been followed-up for at least 2 years after index CBD stone removal and had at least one of the high-risk factors for CBD stone recurrence were included in this study. High-risk factors for CBD stone recurrence were as follows: maximum CBD diameter > 15 mm, CBD angle ≤ 145°, and the presence of periampullary diverticulum (PAD) [12]. Patients included in this study had undergone cholecystectomy at the time of index ERCP since they presented with both GB and CBD stones. These patients were divided up into two groups according to the history of CBD stone recurrence. Patients who belonged to the recurrence group were those who had at least 2 episodes of recurrences during the follow-up period. The non-recurrence group was defined as those who had no recurrence event during the follow-up period. A recurred CBD stone was defined as a particle with a diameter greater than 3 mm which took on the shape of a stone. CBD stones that were found and removed within 6 months after index ERCP were not considered as recurrent stones but as residual stones due to incomplete CBD stone clearance. Only those detected after 6 months following endoscopic CBD stone removal were regarded as recurrent stones [3]. Of the aforementioned patients, 20 who agreed to undergo surveillance endoscopic retrograde cholangiopancreatography (ERCP) procedures to check for CBD stone recurrence were enrolled. All patients were asymptomatic at the time of surveillance ERCP without clinical evidence of biliary tract infection, and had not taken antibiotics nor proton pump inhibitors for at least 3 months prior to bile sample collection.

From these 20 subjects, a total of 20 bile samples (11 from the CBD stone recurrence group and 9 from the non-recurrence group) were collected during scheduled surveillance ERCP from September 2018 to August 2019. After advancing the side-viewing endoscopes (TJF240/JF-260V; Olympus Optical, Tokyo, Japan) into the duodenum 2nd portion and facing the papilla enface, the cannulation catheter was passed through the working channel and inserted into the CBD through the manipulated ampulla. Two to five milliliters of the patients’ bile aspirates from bile duct were sucked out via the catheter before the injection of contrast agent for ERCP procedure. A portion of the sample was stored at −80 °C for NGS analysis and the remainder was stored at room temperature for analysis of bile juice components. Afterwards, dye was injected to evaluate for the presence of CBD stones and removed when present. CBD stones were broadly classified as being cholesterol or pigment stones (black or brown) based on their characteristic external appearance by gross inspection [13], and all recurred CBD stones were brown pigment stones.

This study was approved by the Institutional Review Board of Korea University Ansan Hospital (No. 2018AS0208) and the research was conducted in accordance with the Declaration of Helsinki. All patients provided written informed consent upon enrollment.

### 2.2. Next-Generation Sequencing of Bacterial 16s rRNA Fragments in Bile Samples

To analyze the bacterial community, the V3/V4 regions of the 16S rRNA gene of DNA extracted from the bile samples were amplified, sequenced, and analyzed. The extraction method for bacterial DNA was performed by using a PowerMax Soil DNA Isolation Kit (MO BIO). Each sequenced sample was prepared according to the Illumina 16S Metagenomic Sequencing Library protocols to amplify the V3 and V4 regions (519F-806R). The DNA quality was measured by PicoGreen and NanoDrop. Input genomic DNA (10 ng) was PCR amplified. The barcoded fusion primer sequences used for amplifications were as follows: 341F: 5′ CCTACGGGNGGCWGCAG 3′, and 806R: 5′ GACTACHVGGGTATCTAATCC 3′ containing forward overhang adapter pair. The final purified product was then quantified using qPCR according to the qPCR Quantification Protocol Guide (KAPA Library Quantification kits for Illumina Sequencing platforms) and qualified using the LabChip GX HT DNA High Sensitivity Kit (PerkinElmer, MA, USA). Then, the paired-end (2 × 300 bp) sequencing was performed by the Macrogen using the MiSeq™ platform (Illumina, San Diego, CA, USA).

The operational taxonomic units (OTUs) were generated to calculate the number of microbial species in the bile. In order to compare the bacterial community richness of bile between the two groups, observed OTU-based analyses (OTU richness and Chao 1) were performed. OTU richness is the number of bacterial species observed in each individual sample. Chao 1 represents species richness estimators of the expected OTUs present in a group. As Chao 1 measures OTUs expected in samples, given all the bacterial species that were identified in the samples, Chao1 gives more weight to the low abundance species. To determine α-diversity, the Shannon diversity index and the inverse Simpson index were used. The Shannon diversity index is a quantitative measure that shows how evenly the basic individuals are distributed by taking into account and reflecting the number of different species in each sample; a high Shannon diversity index signifies evenness, whereas a low Shannon diversity index denotes unevenness. The Simpson index measures the degree of concentration or dominance of certain species in a sample. Therefore, the inverse Simpson index is an indicator of how evenly the species are distributed; a high inverse Simpson index implies evenness or a lack of dominance, and a low inverse Simpson index suggests unevenness or the presence of dominance. Evaluation of β-diversity, which analyzes community similarity, was performed by calculating pairwise distances using the phylogenetic metric UniFrac.

### 2.3. Composition of Bile

The bile samples were analyzed for composition of bile, including electrolyte, total bilirubin, total bile acid, cholesterol, phosphorus and Ca2+. Total bilirubin concentration was measured in aliquot on a Cobas 8000 C702 (Roche Diagnostics System, Switzerland). pH, sodium, potassium, chloride, phosphorus, and calcium estimations were carried out on a cobas 8000 ISE (Roche Diagnostics System, Switzerland) using an ion-selective electrode method. For measuring cholesterol and triglycerides, bile was diluted one to five times with methanol and the protein precipitate was removed by centrifugation. The methanol was then evaporated and the residue was taken for analysis. The cholesterol and triglycerides were analyzed by photometric analysis on a Cobas 8000 C702 (Roche Diagnostics System, Switzerland). Total bile acids were also measured with an enzymatic method on a Roche cobas 8000 C702 analyser.

### 2.4. Statistical Analysis

Statistical analyses were performed using IBM SPSS Statistics version 20.0 (IBM, Armonk, NY, USA). The continuous variables were compared using the Student’s t-test. Fisher’s exact test was used to analyze the categorical variables. The sequences generated from pyrosequencing were analyzed with variable software for pre-processing (quality-adjustment, barcode split), identification of OTUs, taxonomic assignment, community comparison, and statistical analysis. The FASTP program was used to remove the adapter sequence, and error-correction was performed for areas where the two reads overlap. The paired-end data, separated by each sample, was assembled in one sequence using FLASH (v. 1.2.11). To ensure that any subsequent analysis was highly accurate, sequences shorter than 400 bp were discarded. By using CD-HIT-OTU, after removing the low-quality sequence, ambiguous reads, chimera reads, etc., which are based on errors, the filtered reads were clustered by identity as OTUs at 97% similarity. The representative sequence of each OTU was performed by BALSTN (v. 2.4.0) to the NCBI 16S Microbial DB, performing a taxonomic assignment with the organism information of the most similar subject. The taxonomy is not assigned if the query coverage and identity score matched to the reference are less than 85%. Community diversity was estimated by using the Shannon and inversed Simpson indexes. The weighted UniFrac distance method was used to perform a principal coordinates analysis (PCoA), and trees were built by the unweighted-pair group method with arithmetic mean (UPGMA).

## 3. Results

### 3.1. Baseline Characteristics

The mean age of the patients was 62.3 ± 16.6 years (range, 29 to 83 years) and 9 (45.0%) patients were female. The anatomical risk factors for recurrent CBD stones, including CBD diameter, presence of PAD, and angulation of the bile duct ≤ 145°, were not statistically different between the two groups (Table 1).

### 3.2. Richness of Microbiota in Bile

A total of 303 bacterial species were identified from 20 bile samples. The average OTU, obtained by calculating the mean values after counting the number of observed OTUs in each sample, was reduced in the 11 individuals with CBD stone recurrence compared to the 9 with non-recurrence (29.6 ± 3.6 vs. 60.4 ± 11.6, *p* = 0.0005, Wilcoxon rank-sum test; Figure 1a). The expected total number of OTUs for each group, as estimated by the Chao1 estimator, was also considerably lower in the recurrence group than in the non-recurrence group (29.8 ± 3.5 vs. 60.5 ± 11.6, *p* = 0.005, Wilcoxon rank-sum test; Figure 1b).

### 3.3. Diversities of Microbiota in Bile

The α-diversities of bile samples, as measured by the Shannon diversity index and the inverse Simpson index, were significantly lower in the recurrence group than those of the non-recurrence group samples. The Shannon index, which reflects equal microbial proportional abundance in a species, was lower in the recurrence group compared to the non-recurrence group (0.65 ± 0.36 vs. 3.12 ± 0.45, *p* = 0.027, Figure 2a), signifying that microbial proportional abundance was uneven in the recurrence group. The inverse Simpson index, which is another way of quantifying the evenness of a species, was also lower in the recurrence group compared to the non-recurrence group (0.46 ± 0.10 vs. 0.75 ± 0.06, *p* = 0.036, Figure 2b), meaning that certain species were more dominant than other species in the recurrence group. According to the Shannon index and inverse Simpson index, the diversity index was statistically higher in the bile of the non-recurrence group than that obtained in the patients with recurrent CBD stones.

### 3.4. Microbial Community Similarity

The unweighted UniFrac distance metric, which measures the similarity in the microbial communities of the bile samples (β-diversity), revealed an overall microbial composition difference between the two groups (unweighted UniFrac, PERMANOVAR, pseudo-F: 1.655, *p* = 0.05, Figure 3). This result indicates that the microbiome clustering in the recurrence group was different from that of the non-recurrence group.

### 3.5. Microbiome

At the phylum level, Bacteroidetes spp. were significantly decreased in the recurrence group, whereas Actinobacteria and Firmicutes spp. were over-represented in the recurrence group relative to the non-recurrence group (Table 2). At the genus level, three bacterial taxa displayed significantly different abundance between the recurrence group and the non-recurrence group. The three genera, *Neisseria*, *Capnocytophaga*, and *Gemella*, were enriched in the non-recurrence group. Conversely, *E. coli*, *Enterococcus* spp., *Klebsiella* spp., *Acinetobacter* spp., *Streptococcus* spp., and *Staphylococcus*, which exert β-glucuronidase activity, were more frequent in the recurrence group, even though there was no statistical difference (Table 3).

### 3.6. Bile Composition

Bile acid concentration was higher in the non-recurrence group than in the recurrence group (254.51 ± 82.0 mmol/L vs. 147.1 ± 64.2 mmol/L, *p* < 0.01). Neither the pH values nor Ca^2+^ showed statistical differences between the bile samples of both groups (Table 4).

## 4. Discussion

In the present study, 16S rRNA gene profiling analysis using NGS allowed us to observe the difference in bile microbiota composition between the CBD stone recurrence group and the non-recurrence group. Although the sample size (N = 20) might not be large enough to generalize the results, the two groups significantly differed in terms of richness and evenness in this preliminary study. The number of species was decreased and the dominance of certain species was present in the bile samples of the recurrence group. On the other hand, all types of species were equally abundant, with no specific species being predominant in the bile samples of the non-recurrence group. In addition, the difference in species composition and clustering between the two groups were also shown.

Recent progress in understanding the symbiosis of human microbiota revealed an important role in immune homeostasis [14,15]. A balance of these commensal microbial species is probably required to prevent infection with pathogens and pathological inflammation. However, this dynamic equilibrium can be altered at any time by environmental factors and external interferences. These microbial alterations and imbalances with decreased richness and diversity in the pathologic condition group is a phenomenon called dysbiosis [16]. There is increasing evidence that the dysbiosis of a microbiome is linked to many gastrointestinal and systemic diseases. An experimental study reported that chemical- and pathogen-induced intestinal inflammation resulted in the loss of microbial density and diversity and led to the proliferation of Gram-negative bacteria which possess β-glucuronidase activity in the intestine [17]. A balance of the microbial species in bile is probably also required to prevent infection with pathogens and pathological inflammation related to the formation of gallstones [10]. Thus, it could be speculated that bacterial dysbiosis with the loss of microbial richness and diversity in bile could lead to inflammation of the bile duct, in favor of biofilm formation with resultant proliferation of bacteria with β-glucuronidase activity. In fact, a notable dysbiosis of the bile microbiome, including decreased richness and diversity in the bile of the recurrence group, was found in the current study. Furthermore, the finding that *E. coli*, *Enterococcus* spp., *Klebsiella* spp., *Pseudomonas* spp., *Acinetobacter* spp., *Streptococcus* spp., and *Staphylococcus*, which exert β-glucuronidase activity, were more frequent in the recurrence group additionally supports the validity of the speculation of this study.

Whereas the abovementioned bacteria were more frequently found in the recurrence group, bacteria in the genera *Neisseria*, *Gemella*, and *Capnocytophaga* were expressed more frequently in the non-recurrence group. All of these three bacteria are part of the normal oral flora associated with periodontitis and dental plaque [18]. There is very little information about the possible clinical significance on the protective role of these bacteria in the formation of biliary stones. Further studies using molecular techniques will be needed and may provide new insights into the pathophysiology of bile duct stone formation and the involvement of microbes or their metabolites in this process.

In addition to differences in the microbiome, the bile component analysis also showed that the two groups had different metabolic profiles with respect to bile acid. Bile acids act as detergents and have an important role in solubilizing dietary lipids and fat-soluble vitamins to facilitate their absorption in the small intestine [19]. Previous studies have shown that decreased concentration of bile salts in the bile diminishes the micellar solubilization of bilirubin, as well as cholesterol, favoring the formation of brown pigment stones [20,21]. In addition to their role in the digestion of lipids, bile acids generally inhibit bacterial growth in the small intestine as a major regulator of the gut microbiota, and prevent retrograde bacterial infections in the CBD [22]. Bile acids themselves can also modulate the composition of the microbiota in the bile duct, directly or indirectly, through the activation of the innate immune system [23,24]. Inflammation in the bile duct could induce oxidative stress in harmful microbiomes, which is related to the formation of gallstones through specific enzymatic activities or the production of biofilm [25,26]. Conversely, other studies have shown that gut microbiota has profound effects on bile acid metabolism and secondary bile acid production [27,28]. The alteration of gut microbiome could change the composition of the bile acid pool with the regulation of secondary bile acid metabolism and inhibition of bile acid synthesis by alleviating FXR inhibition in the ileum [28,29]. Since it seems to be a chicken-and-egg question, to further explore the bile acid functionality, study on the bile acid-bacteria-host interplay at the pathway level would be necessary.

The strength of this study is that microbiological identification of unculturable or difficult to culture bacteria was enhanced using recent advancements in NGS. However, there have been few extensive NGS-based research studies on bile samples. While a recent study focused on the relationships among gut microbiota, bile acids, and cytokines in blood and stool samples of patients with cholangiocarcinoma [30], our study revealed the complexity and specificity of the biliary microecology and metabolites in bile samples of patients with recurrent CBD stones. Therefore, in addition to gaining a better understanding of the disease processes of pathogens, the knowledge gained from this bile research may ultimately lead to the design of new antimicrobial treatments or assist in the development of improved probiotic strains and supplementation of bile acid composition in the future.

Although the abovementioned investigations attempted to provide a comprehensive insight into the potential contribution of the microbiome and metabolic component in bile related to CBD stone recurrence, several limitations need to be addressed. First, aspirated bile samples contain floating bacterial species but cannot detect sessile bacterial populations adhering to the mucosa or those residing in the biofilm. However, since a study demonstrated that bacteria in the bile fluid almost always coincided with the presence of bacteria on the bile duct wall [31], bacteria species floating in bile may sufficiently reflect the total microbial environment of biliary tract. Nevertheless, additional bile duct tissue biopsies can be expected to reveal more accurate microbial communities in the CBD. Second, despite the fact that this study elucidated aspects of the richness and diversity of the biliary microbiota related to recurrent CBD stones by 16S sequencing, a more comprehensive understanding of biliary bacterial function and the microbe–host interaction would require further investigation by methods such as whole-metagenome shotgun sequencing, metatranscriptomic, metametabolomic, and metaproteomic technologies. Third, even though a specific profile of bile salts and their concentrations appear to be key factors in lithogenic potency, exploration on the composition of individual bile salts in the bile was not performed in this study.

In conclusion, polymicrobial infections were seen in all study subjects who underwent ampullary manipulations during ERCP. The patients with recurrent CBD stones showed microbial dysbiosis with a significant reduction in microbial richness and diversity in the bile. The bile composition was also dissimilar between the two groups. A significant difference in the concentration of bile acid was also found between the recurrence group and non-recurrence group. Although the results of this study are preliminary in nature and require confirmation, microbiota dysbiosis and altered bacterial community assembly in bile duct, and decreased bile acid, were associated with recurrence of bile duct stones.

## Figures and Tables

**Figure 1 jcm-10-03303-f001:**
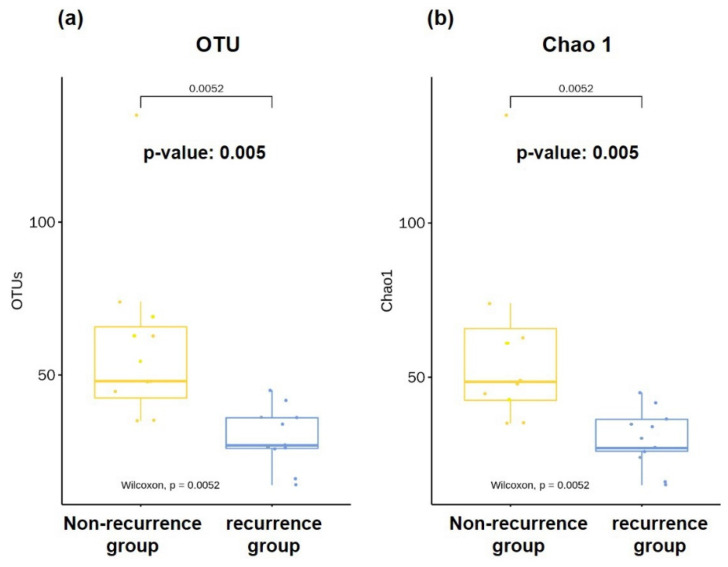
Comparison on richness of microbiota in bile. Microbiota richness, represented as the number of (**a**) observed operational taxonomic units (OTU) and (**b**) Chao 1, was reduced in the recurrent CBD stone group (*n* = 11) compared with the non-recurrence group (*n* = 9).

**Figure 2 jcm-10-03303-f002:**
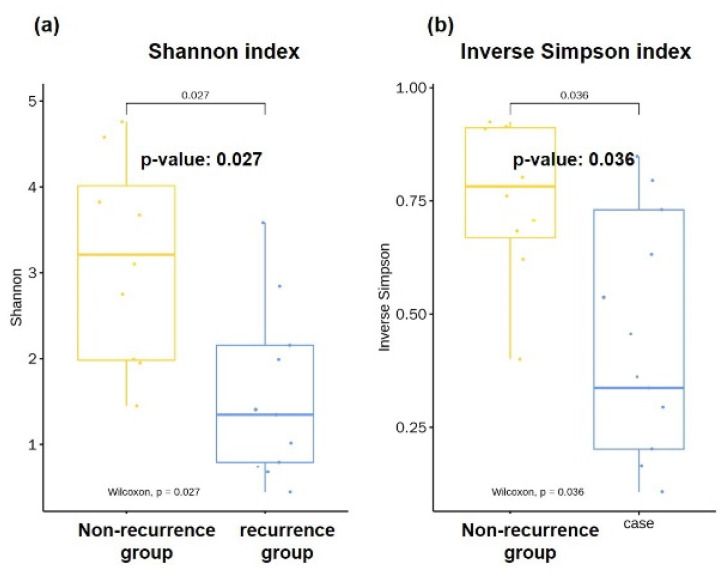
Comparison on evenness and dominance of microbiota in bile. Mean α-diversity, measured by (**a**) Shannon index and (**b**) inverse Simpson index, was reduced in the CBD stone recurrence group compared to the non-recurrence group.

**Figure 3 jcm-10-03303-f003:**
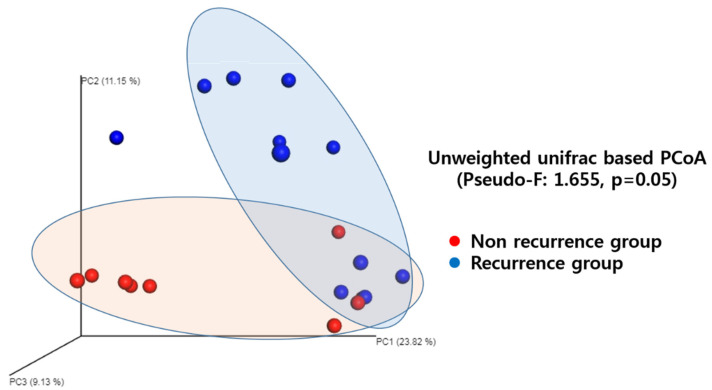
Comparison on community similarity. The unweighted uniFrac distance metric shows difference in microbial communities between the non-recurrence group and recurrence group.

**Table 1 jcm-10-03303-t001:** Baseline characteristics.

	Recurrence Group (N = 11)	Non-Recurrence Group (N = 9)	*p*-Value
Age, median years (min-max)	62 (32–78)	63 (29–83)	0.12
Gender, male (%)	6 (66.6)	5 (62.5)	0.45
BMI, median kg/m^2^(min-max)	22.6 (17.6–29.0)	22.5 (16.2–29.4)	0.45
CBD diameter (mm, mean ± SD)	17.7 ± 2.5	17.8 ± 2.2	0.89
Angulation of bile duct (≤145°) (N, %)	1 (11.1)	0	0.85
PAD (N, %)			0.20
I	2	1	
II	5	4	
III	2	3	

CBD, common bile duct; BMI, body mass index; PAD, periampullary diverticulum.

**Table 2 jcm-10-03303-t002:** List and relative proportion of phylum identified by next-generation sequencing.

Phylum	Recurrence Group		Non-Recurrence Group		*p*-Value
	Mean Proportion	SEM	Mean Proportion	SEM	
Euryarchaeota	<0.001	<0.001	0.001	0.001	0.345
Actinobacteria	0.024	0.005	0.009	0.003	0.048
Bacteroidetes	0.074	0.045	0.188	0.066	0.042
Chloroflexi	<0.001	<0.001	0.002	0.003	0.345
Cyanobacteria	0.001	0.002	<0.001	<0.001	0.084
Deinococcus	<0.001	<0.001	0.001	0.001	0.346
Firmicutes	0.384	0.105	0.178	0.027	0.038
Fusobacteria	0.009	0.006	0.021	0.017	0.311
Gemmatimonadetes	<0.001	<0.001	0.001	0.001	0.345
Proteobacteria	0.504	0.121	0.568	0.094	0.814
Spirochaetes	<0.001	<0.001	0.001	0.001	0.345
Synergistetes	<0.001	<0.001	0.008	0.008	0.409
Tenericutes	<0.001	<0.001	<0.001	<0.001	0.999
Unassigned	0.001	<0.001	0.017	0.013	0.411

SEM, standard error of the mean.

**Table 3 jcm-10-03303-t003:** List and relative proportion of genus identified by next-generation sequencing.

Genus	Recurrence Group		Non-Recurrence Group		*p*-Value
	Mean Proportion	SEM	Mean Proportion	SEM	
*Neisseria*	0.001	0.001	0.162	0.021	0.008
*Capnocytophaga*	0.001	0.001	0.137	0.004	0.025
*Gemella*	<0.001	<0.001	0.075	0.003	0.041
*Rothia*	<0.001	<0.001	0.003	0.001	0.052
*Haemophilus*	0.041	0.011	0.046	0.033	0.075
*Streptococcus*	0.132	0.102	0.029	0.009	0.091
*Klebsiella*	0.134	0.049	0.049	0.026	0.162
*Enterococcus*	0.033	0.006	0.006	0.003	0.191
*Clostridium*	0.044	0.029	0.059	0.029	0.201
*Pseudomonas*	0.081	0.023	0.073	0.048	0.746
*Staphylococcus*	0.003	0.002	0.002	0.001	0.807
*Acinetobacter*	0.002	0.001	0.001	0.001	0.957
*Lactobacillus*	0.034	0.034	0.001	0.001	0.999
*Citrobacter*	0.163	0.126	0.014	0.009	0.999
*Escherichia*	0.320	0.123	0.282	0.107	0.999
*Aeromonas*	0.001	0.001	0.007	0.006	0.473
*Fusobacterium*	0.009	0.007	0.028	0.018	0.245
Unassigned	0.001	<0.001	0.017	0.013	0.411

SEM, standard error of the mean.

**Table 4 jcm-10-03303-t004:** Bile metabolic profile.

	Recurrence Group (N = 11)	Non-Recurrence Group (N = 9)	*p*-Value
pH	7.8 ± 0.2	7.95 ± 0.4	0.85
Total nucleated cell count	12 ± 5	22 ± 10.4	0.07
Total bilirubin (mg/dL)	37.5 ± 63.5	27.7 ± 80.7	0.85
Cholesterol (mg/dL)	28.1 ± 50.7	34.2 ± 62.1	0.32
Bile acid (mmol/L)	147.1 ± 64.2	254.5 ± 82.0	<0.01
Phospholipid (mg/dL)	324 ± 272	454 ± 350	0.25
Ca^2+^ (mg/dL)	7.7 ± 4.4	5.5 ± 2.2	0.65

CBD, common bile duct.

## Data Availability

The data presented in this study are available on request from the corresponding author.

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
