# Peer review of "Analysis on Microbial Profiles & Components of Bile in Patients with Recurrent CBD Stones after Endoscopic CBD Stone Removal: A Preliminary Study"

_jcm, 2021, doi:10.3390/jcm10153303_

Round 1
Reviewer 1 Report
The article by J.W. Choe et al reviews the profile of the microbiota and the different components of the bile in patients with recurrent CBD stones compared to patients who overcame and did not relapse after curative endoscopy.
Although the sample size is small (11 vs 9) patients, the article is of interest, since very little is known about the role of the microbiota in the composition of bile acids in different pathologies and viceversa.
The manuscript is well written, the introduction focuses well on the topic of interest, the objectives are clear and the methodology used is current and in accordance with the results obtained. The discussion summarizes the main results and compares them with the existing bibliography
- The methodology for determining the composition of bile should be developed a little more or refer to the bibliography
- Is the bile acid concentration found in the two groups of patients similar to that found in a healthy subject?
- Perhaps the authors should take into consideration the concentration of bile acids in physiological situations and in other pathologies. It might be helpful if the authors refer to the following article PMID: 31298745 DOI: 10.1002 / hep.30852
Author Response
Dear Editor-in-Chief of JCM:
Thank you for your helpful comments on the manuscript titled “Analysis on microbial profiles & components of bile in patients with recurrent CBD stones after endoscopic CBD stone removal (jcm-1282410).” We have revised our manuscript as suggested by the reviewers and agree to the points the reviewers have indicated. They are as follows:
[Reviewer’s Comments]
Reviewer 1
Major concerns:
- The methodology for determining the composition of bile should be developed a little more or refer to the bibliography
--> As suggested, the following paragraph was added in the Methods section
“The bile samples were analyzed for composition of bile, including electrolyte, total bilirubin, total bile acid, cholesterol, phosphorus and Ca2+. Total bilirubin concentration was measured in aliquot on a Cobas 8000 C702 (Roche Diagnostics System, Switzerland). pH, sodium, potassium, chloride, phosphorus and calcium estimations were done on cobas 8000 ISE (Roche Diagnostics System, Switzerland) using ion-selective electrode method. For measuring cholesterol, triglycerides, bile was diluted one to five times with methanol and the protein precipitate was removed by centrifugation. The methanol was then evaporated and residue was taken for analysis. The cholesterol and triglycerides were analyzed by photometric analysis on Cobas 8000 C702 (Roche Diagnostics System, Switzerland). Total bile acids also measured with an enzymatic method on the Roche cobas 8000 C702 analyser.”
- Is the bile acid concentration found in the two groups of patients similar to that found in a healthy subject?
--> According to the review by Hofmann,1 It is reported that “The concentration of bile acids at any place in enterohepatic circulation ranges from 20 to 300 mmol/L, depending on the relation of the rates of input of bile acids and surrounding aqueous fluid”.
Reference
- Hofmann AF. The continuing importance of bile acids in liver and intestinal disease. Archives of Internal Medicine 1999;159(22):2647-58. doi: DOI 10.1001/archinte.159.22.2647
--> In our study, the concentration of bile acid found in the two groups (147.1 ± 64.2 mmol/L and 254.5 ± 82.0 mmol/L) tended to be high, but was within normal range. This might be due to the fact that all patients had at least one of the high-risk factors for CBD stone recurrence (maximum CBD diameter >15 mm, CBD angle ≤145°, and the presence of periampullary diverticulum), which are conditions that cause bile stasis leading to concentration of bile acid.
- Perhaps the authors should take into consideration the concentration of bile acids in physiological situations and in other pathologies. It might be helpful if the authors refer to the following article PMID: 31298745 DOI: 10.1002 / hep.30852
--> Thank you for providing us with additional important study related to the current topic of our research. We have reviewed and included pertinent passage from the paper you have mentioned in the discussion section as shown below.
Before: This study revealed the complexity and specificity of the biliary microecology and metabolites in bile samples
After: While a recent study had focused on the relationships among gut microbiota, bile acids, and cytokines in blood and stool samples of patients with cholangiocarcinoma,35 our study revealed the complexity and specificity of the biliary microecology and metabolites in bile samples of patient with recurrent CBD stones.
Reviewer 2
The main flaw of this study is that, in the face of the considerable complexity of the technologically advanced analysis of the microbial components of bile, the number of the two groups of patients examined is very low and this greatly limits the validity of the statistical analysis. What conclusions can be drawn with a control group of only 9 patients? The control group should have been at least double the number of cases. Authors should strongly emphasize this possible bias at the beginning of the discussion.
--> We have added a paragraph at the beginning in Discussion section to help the readers understand the current limitation of small sample size and also the potential possibility of bias as you have mentioned.
Before: Two groups significantly differed in terms of richness and evenness.
After: Although the sample size (N=20) might not be large enough to generalize the results, two groups significantly differed in terms of richness and evenness.
There are also many minor points that still need to be corrected.
1) in line 98 on page 3 it is said that the bile duct stones were roughly classified as cholesterol stones or pigment stones but then in line 100 it says that all the recurring stones were brown pigment stones.
--> Pigment stones are classified as either brown or black pigment stones. When there is recurrence, it is almost always brown pigment stone and we intended to clarify this in the Materials and Methods section to help the readers understand this factual finding. For clarity reasons, we have added the types of pigment stone in the parenthesis as shown below.
Before: CBD stones were broadly classified as being cholesterol or pigment stones based on their characteristic external appearance by gross inspection,[13] and all recurred CBD stones were brown pigment stones.
After: CBD stones were broadly classified as being cholesterol or pigment stones (black or brown) based on their characteristic external appearance by gross inspection,[13] and all recurred CBD stones were brown pigment stones.
2) Methods for studying the components of bile should be described in much more detail
--> As suggested, the following paragraph was added in the Method section
“The bile samples were analyzed for composition of bile, including electrolyte, total bilirubin, total bile acid, cholesterol, phosphorus and Ca2+. Total bilirubin concentration was measured in aliquot on a Cobas 8000 C702 (Roche Diagnostics System, Switzerland). pH, sodium, potassium, chloride, phosphorus and calcium estimations were done on cobas 8000 ISE (Roche Diagnostics System, Switzerland) using ion-selective electrode method. For measuring cholesterol, triglycerides, bile was diluted one to five times with methanol and the protein precipitate was removed by centrifugation. The methanol was then evaporated and residue was taken for analysis. The cholesterol and triglycerides were analyzed by photometric analysis on Cobas 8000 C702 (Roche Diagnostics System, Switzerland). Total bile acids also measured with an enzymatic method on the Roche cobas 8000 C702 analyser.”
3) The discussion appears to be too long and verbose. Surely it should be reduced by at least 50%
--> Thank you for helpful comment. We have shortened the discussion section as suggested.
4) The percentage composition of the individual bile salts and not just the total amount of the bile salts should be reported
--> As you have suggested, the percentage composition of the individual bile salts should have been reported, but only the total amount of the bile salts was reported in the current study. This is one of the limitations of current study and has been acknowledged in the Discussion Section as shown below.
“Third, even though specific profile of bile acids and their concentrations appear to be key factors in lithogenic potency, exploration of the bile proteome was not performed in this study.”
5) It is not at all clear how the results of this study can be translated into clinical practice
--> It would be best if the results of this study could be translated into clinical practice. Thus, we have mentioned in the Discussion Section about the potential clinical application as shown below.
“Therefore, in addition to gaining a better understanding of the disease processes of pathogens, the knowledge gained from this bile research would ultimately lead to the design of new antimicrobial treatments or assist in the development of improved probiotic strains and supplementation of the bile acid composition in the future.”
6) In the methods it is not reported whether the patients were taking proton pump inhibitor drugs which, by raising the gastric pH, can interfere with the duodenal bacterial flora and therefore also on the biliary one
--> Thank you for helpful comment. We totally agree with your opinion that PPIs could influence the gut microbiota related to bile microbiologic environment. There were no PPI users for 3 months before scheduled ERCP in both groups in our study. Therefore, we added the comment on “PPI use” in the Methods section as shown below.
Before: All patients were asymptomatic at the time of surveillance ERCP without clinical evidence of biliary tract infection, and had not taken antibiotics for at least 3 months prior to bile sample collection
After: All patients were asymptomatic at the time of surveillance ERCP without clinical evidence of biliary tract infection, and had not taken antibiotics and proton pump inhibitors for at least 3 months prior to bile sample collection.
7) how can the authors be sure that there has been no contamination of the catheter with which the bile was taken from the bile duct?
--> There is always a possibility that contamination could have occurred. However, to minimize contamination, sterile saline was injected through the working channel before catheter insertion. While aspirating the bile from the bile duct, bile was aspirated at the beginning of the procedure using a new cannulation catheter or the catheter that had been sterilized prior to use.
We agree with the reviewers in all the points and the corrections in an annotated version are the points the reviewers have indicated. Thank you and the reviewers again for considering our manuscript to be published in Journal of Clinical Medicine. We look forward to receiving your answer soon.
Sincerely,
Hong Sik Lee, M.D., Ph.D.
Division of Gastroenterology and Hepatology, Department of Internal Medicine,
Korea University Anam Hospital, Korea University College of Medicine,
73 Inchon-ro, Seongbuk-gu, Seoul, 02841, Korea
Tel: +82-2-920-6555, Fax: +82-2-953-1943,
E-mail: hslee60@korea.ac.kr

Reviewer 2 Report
The paper by Jung Wan Choe and collaborators is aimed to establish whether the microbiological profile and bile composition of patients with recurrent main biliary tract stones are different from the bile profile of those who have not had recurrence episodes.
The main flaw of this study is that, in the face of the considerable complexity of the technologically advanced analysis of the microbial components of bile, the number of the two groups of patients examined is very low and this greatly limits the validity of the statistical analysis. What conclusions can be drawn with a control group of only 9 patients? The control group should have been at least double the number of cases. Authors should strongly emphasize this possible bias at the beginning of the discussion.
There are also many minor points that still need to be corrected.
1) in line 98 on page 3 it is said that the bile duct stones were roughly classified as cholesterol stones or pigment stones but then in line 100 it says that all the recurring stones were brown pigment stones.
2) Methods for studying the components of bile should be described in much more detail
3) The discussion appears to be too long and verbose. Surely it should be reduced by at least 50%
4) The percentage composition of the individual bile salts and not just the total amount of the bile salts should be reported
5) It is not at all clear how the results of this study can be translated into clinical practice
6) In the methods it is not reported whether the patients were taking proton pump inhibitor drugs which, by raising the gastric pH, can interfere with the duodenal bacterial flora and therefore also on the biliary one
7) how can the authors be sure that there has been no contamination of the catheter with which the bile was taken from the bile duct?
Author Response

(The authors gave the same response as above.)

Round 2
Reviewer 2 Report
"a preliminary study" should be reported in the abstract title.
The composition of the bile in the individual bile salts has not been reported.
Author Response
[Reviewer’s Comments]
- "a preliminary study" should be reported in the abstract title.
--> As suggested, the following paragraph was modified in the Abstract
Before: The aim of this study was to investigate the microbial profile and components of bile in patients with and without recurrent CBD stones after endoscopic CBD stone removal.
After: The aim of this preliminary study was to investigate the microbial profile and components of bile in patients with and without recurrent CBD stones after endoscopic CBD stone removal.
- The composition of the bile in the individual bile salts has not been reported.
--> As you have suggested, the percentage composition of the individual bile salts should have been reported, but only the total amount of the bile salts was measured in the current study. In the future studies, we are planning on analyzing the composition of the individual bile salts. As suggested, the following paragraph was modified in the Discussion section.
Before: Third, even though specific profile of bile acids and their concentrations appear to be key factors in lithogenic potency, exploration of the bile proteome was not performed in this study.”
After: Third, even though specific profile of bile salts and their concentrations appear to be key factors in lithogenic potency, exploration on the composition of individual bile salts in the bile was not performed in this study.
Sincerely,
Hong Sik Lee, M.D., Ph.D.
